# Accelerating Payload Release from Complex Coacervates through Mechanical Stimulation

**DOI:** 10.3390/polym15030586

**Published:** 2023-01-23

**Authors:** Wesam A. Hatem, Yakov Lapitsky

**Affiliations:** Department of Chemical Engineering, University of Toledo, Toledo, OH 43606, USA

**Keywords:** polyelectrolyte, complex coacervate, polyamine, stimulus-responsive materials, controlled release

## Abstract

Complex coacervates formed through the association of charged polymers with oppositely charged species are often investigated for controlled release applications and can provide highly sustained (multi-day, -week or -month) release of both small-molecule and macromolecular actives. This release, however, can sometimes be too slow to deliver the active molecules in the doses needed to achieve the desired effect. Here, we explore how the slow release of small molecules from coacervate matrices can be accelerated through mechanical stimulation. Using coacervates formed through the association of poly(allylamine hydrochloride) (PAH) with pentavalent tripolyphosphate (TPP) ions and Rhodamine B dye as the model coacervate and payload, we demonstrate that slow payload release from complex coacervates can be accelerated severalfold through mechanical stimulation (akin to flavor release from a chewed piece of gum). The stimulation leading to this effect can be readily achieved through either perforation (with needles) or compression of the coacervates and, besides accelerating the release, can result in a deswelling of the coacervate phases. The mechanical activation effect evidently reflects the rupture and collapse of solvent-filled pores, which form due to osmotic swelling of the solute-charged coacervate pellets and is most pronounced in release media that favor swelling. This stimulation effect is therefore strong in deionized water (where the swelling is substantial) and only subtle and shorter-lived in phosphate buffered saline (where the PAH/TPP coacervate swelling is inhibited). Taken together, these findings suggest that mechanical activation could be useful in extending the complex coacervate matrix efficacy in highly sustained release applications where the slowly releasing coacervate-based sustained release vehicles undergo significant osmotic swelling.

## 1. Introduction

Complex coacervation is a liquid-liquid phase separation that occurs through the complexation of colloidal (or macromolecular) solutes with other solute species [1,2,3]. This phase separation generates a solute-rich coacervate phase, which is rich in both the associating solution species and in equilibrium with a dilute, solvent-rich supernatant phase. The colloid-rich coacervate phase typically has viscoelastic fluid- or gel-like properties [4,5,6], and offers numerous benefits: easy formation under mild, aqueous conditions [7,8]; low toxicity [9,10]; and an ability to form, transform their properties, and dissolve in response to external stimuli [1,11,12,13]. Among their many potential applications (which range from drug delivery [8,14,15] to separation processes [16,17], foods [18], and adhesives [11,19,20]), complex coacervates are frequently used in the controlled release of various active compounds [8,14,15,21,22,23].

One aspect of complex coacervates that makes them potentially effective for sustained release applications is their polymer-rich composition, which—in contrast to most hydrogels [24]—makes them highly effective diffusion barriers [9,15,25,26]. This barrier property enables them to sustain the release of small water-soluble molecules (which tend to elute rapidly from gels) over highly extended timescales. Complex coacervates formed from poly(allylamine hydrochloride) (PAH) complexed with the multivalent anion tripolyphosphate (TPP), for instance, have recently been shown to sustain the release of diverse small molecules (including drugs and disinfectants) over multiple months [9,15,21,27].

While the highly sustained release enabled by these viscoelastic materials could be useful in an array of biomedical, household, and industrial technologies, the duration of the benefits derived from such release can sometimes be limited by the payload delivery ultimately becoming too slow to be efficacious (once the most-accessible portion of their payload closest to the surface is released) [21,27]. A recent study, for instance, revealed that, while the long-term bactericide release from PAH/TPP coacervates can provide antibacterial benefits over multiple weeks, this activity is ultimately lost (even though only a fraction of the bactericidal payload is released) [21]. The evident reason for this ultimate activity loss is that the release rate declines with time and, after a few weeks, becomes too slow to be effective. Indeed, with certain coacervate/payload molecule combinations, the slower-than-desired release can impose even greater limitations on the coacervate’s applicability (e.g., to situations/applications where the volumes of the release media relative to the coacervate are very low, such that the released actives are not diluted below the minimal concentration needed for their efficacious use) [27].

To overcome these limitations of insufficient release rates or early activity loss, here we explore the use of mechanical stimulation—namely, periodic perforation and compression—as a method for stimulating or reactivating small molecule release from coacervate matrices. Mechanical crushing of coacervate-based microcapsules (e.g., in ink, fragrance, or food formulations) is a well-known approach to achieving rapid stimulus-responsive release of hydrophobic payloads such as oil droplets [28,29,30,31]. However, we are not aware of any reports demonstrating mechanical stimulus use for controlling (1) long-term (multiday) release from complex coacervates, (2) the release of hydrophilic/water-soluble actives from complex coacervates, or (3) the release from continuous coacervate phases (i.e., macroscopic matrices) rather than dispersed coacervate microcapsules. To this end, using macroscopic PAH/TPP coacervates and the Rhodamine B (RhB) dye as a model complex coacervate and small, hydrophilic payload system, we analyze the effect of mechanical stimulation (using UV-vis spectroscopy) on long-term small molecule release. To gain further insight into these effects, the impacts of mechanical stimulation on coacervate swelling are also analyzed (through gravimetry and digital photography) and related to changes in the release profiles. Finally, we discuss the opportunities offered by (and limitations of) this approach in potential applications of complex coacervates.

## 2. Materials and Methods

### 2.1. Materials

All experiments were conducted using Millipore Direct-Q 3 deionized water (18.2 MΩ cm). The PAH (nominal molecular weight of 150 kDa; supplied as a 40 wt% aqueous solution) was purchased from Nittobo Medical Co. Ltd. (Tokyo, Japan). TPP (≥98% pure) and RhB (≥95% pure) were purchased from Sigma-Aldrich (St. Louis, MO, USA). Phosphate buffered saline (PBS) powder, HCl (12 M), and NaOH pellets (≥97% pure) were purchased from Fischer Scientific (Nazareth, PA, Fair Lawn, NJ and Hampton, NH, respectively). All materials were used as received.

### 2.2. Coacervate Preparation

The coacervates were prepared in 2.0 mL microcentrifuge tubes at room temperature using 0.33 mL of 10 wt% PAH and 0.35 mL of 7.5 wt% TPP, both adjusted to pH 7.0 with 6 M HCl and 6 M NaOH solutions. Small (0.097 mL) aliquots of 16 mg/mL aqueous RhB solution were added to the PAH before adding TPP, whereupon the PAH/RhB solutions were mixed for 10–15 s on a vortex mixer. TPP was then immediately added (to generate a 0.20:1 TPP:PAH amine group molar ratio), shaking vigorously for ~10 s by hand after the single-shot addition. The phase-separating PAH/TPP/RhB mixtures were then centrifuged at 15,000 rpm for 90 min, which yielded single, macroscopic coacervate pellets (3–4 mm in height) at the bottoms of the microcentrifuge tubes with dilute/solvent-rich supernatant phases on the top.

### 2.3. Release Experiments

Upon forming the macroscopic coacervate matrix pellets, the supernatant phases were removed, after which the coacervates were weighed, submerged in 1 mL of release media—either deionized water or 1 × PBS (pH 7.4)—and agitated at 400 rpm and 37 °C using a Benchmark Scientific Multitherm shaker (South Plainfield, NJ, USA). To maintain sink conditions and determine the RhB amounts released, the release medium was collected (after 1 h and every 24 h thereafter) and replaced with fresh medium after rinsing the coacervate and tube with 1 mL of fresh, RhB-free release medium for 3–5 s. The released RhB was then quantified by UV-vis spectroscopy, employing a Varian Cary 50 spectrophotometer (λ = 555 nm; ε = 143 mL mg^−1^ cm^−1^ in deionized water and 192 mL mg^−1^ cm^−1^ in PBS).

To mechanically stimulate the release, two methods were used: perforation and compression. In the first release experiment, this agitation was performed every 3 d, with deionized water as the release medium and the first treatment occurring 3 d into the release process. Here, the perforation procedure was performed manually with a needle (a size 7 cotton darner), which was 47 mm long and 0.69 mm in diameter. During each mechanical stimulation step, the perforated samples were subjected to 5 needle jabs with locations resembling the Number 5 face of a game die (Figure 1a). Conversely, stimulation through coacervate compression was performed by wedging a 6.5 mm wide and 0.9 mm thick spatula (with a U-shaped tip) between the coacervate and centrifuge tube. These spatula insertions were performed one per treatment, and their locations were alternated between the treatment times by first inserting the spatula along the front-side tube wall, then the back-side wall, then the right-side wall, and finally, the left-side wall (Figure 1b). Both the needles and spatulas were allowed to reach the bottoms of the microcentrifuge tubes.

To also examine the effects of mechanical stimulation frequency, another release experiment was conducted, where the stimulation was performed daily (with the first treatment being performed 1 d into the release process), and deionized water again served as the release medium. Since increasing the mechanical stimulation frequency without varying the intensity of each treatment increases the total amount of stimulation performed, we decreased the daily perforation in this experiment to 2 needle jabs (where the insertion locations resembled the Number 2 face on a game die and were rotated by ~90° each day). Lastly, to test the release medium’s effect on the mechanical stimulation efficacy, the release experiment was repeated using 1 × PBS (pH 7.4) as the release medium. Here, the coacervates were stimulated daily with either needle jabs or through compression but using PBS as the swelling/release medium instead of deionized water. In each of the above experiments, release from the mechanically stimulated coacervates was compared with that from stimulation-free controls, and all release conditions were analyzed using at least three replicate samples.

### 2.4. Gravimetric Analysis

To gain insights into the mechanical stimulation effects on the coacervate stability, and possibly correlate the release performance to variations in coacervate swelling, changes in the wet coacervate weight were analyzed by gravimetry each time that the release media was replaced. During each such analysis, the supernatant release media was removed from the microcentrifuge tubes, whereupon each coacervate surface was rinsed with fresh release medium (either deionized water or 1 × PBS). To minimize the risk of unintended mechanical stimulation, the rinse stream was aimed at the microcentrifuge tube walls (rather than the coacervate surface) and allowed to gently flow down the tube sides. Any solvent remaining after the rinse step was then carefully removed with a Kimwipe^TM^ (by aspiring it into the corners of the wipes through capillary action while avoiding direct contact with the coacervate surface) before weighing the coacervate phase. The coacervate weights at each time point were normalized to their initial weight before being contacted with the release medium. These gravimetric analyses were supplemented with digital photography, which supplied further evidence of any changes in coacervate size. Like with the spectroscopic release measurements, all gravimetric measurements were performed using three replicate coacervate samples.

## 3. Results and Discussion

### 3.1. Effect of Mechanical Stimulation

The release into deionized water was highly sensitive to mechanical stimulation (Figure 1a). After rapidly releasing an average of ~2 µg RhB within the first day, the control (unstimulated) coacervates rapidly diminished their release rate to significantly less than 1 µg/d and released only a few µg total RhB after 1 month (blue diamonds in Figure 2a,b). Coacervates that were mechanically stimulated every 3 d, on the other hand (regardless of whether this was done through perforation or compression), produced pulsatile release profiles where, on the day following mechanical stimulation, there was an order of magnitude increase in the measured RhB released (such that ~2–4 µg of RhB were released in the day following the stimulation; see red circles and grey squares in Figure 2a,b). After this initial increase, however, the release rates returned to their near-baseline level. Further evidence of this pulsatility of the mechanically stimulated release process came from the plumes of pink RhB dye that were visually seen rising from the coacervate pellets upon their perforation (especially early in the release process and when the perforation occurred at the center rather than near the edges of the pellet). Though this reactivation effect became weaker with repeated application, it produced a discernable effect over ~1 month (as evident in Figure 2b), and the cumulative effect of the mechanical stimulation steps was a severalfold increase in the cumulative release over the ~ 1-month experiment (Figure 2a).

Besides the release profiles, the mechanical stimulation had a marked impact on the coacervate swelling. The control PAH/TPP coacervate pellets swelled significantly when placed in contact with the deionized water release media (blue diamonds in Figure 3a). This swelling caused the coacervate weight to monotonically increase with time and the coacervate pellets to increase in size and become wispy (see Figure 3bi). Such a response to deionized water was qualitatively consistent with that seen for polyanion/polycation coacervates (complexes of oppositely charged polyelectrolytes) in deionized water [32,33], and bactericide-loaded PAH/TPP coacervates in tap water [21,27], and—together with the continued slow release from these coacervates in Figure 2—suggested that the swollen coacervates were composed of water-rich pores dispersed in a continuous coacervate phase. These pores were likely gradually generated due to (1) the high osmotic pressure created by the encapsulated RhB and (possibly) unassociated PAH and TPP, and (2) the greater permeability of the (much smaller) water molecules through the coacervate phase, which allowed the water from the release medium to fill these pores. The view of the PAH/TPP coacervate being more permeable to water than other solutes (e.g., RhB or any uncomplexed PAH) was supported by the observation that these coacervates do not take very long to dry when exposed to air and was consistent with prior studies on the swelling of hydrogels, which showed the transport of water into swelling polymer networks to be faster than the release of osmotic pressure-causing organic solutes [34,35,36,37].

More importantly, the “porous coacervate phase” interpretation was supported by the effect of mechanical stimulation on the normalized coacervate weight. Each mechanical stimulation step (regardless of whether performed by perforation or compression; grey squares and red circles in Figure 3a and images in Figure 3bii,iii), produced a sudden reduction in swelling, evidently due to the rupture/collapse of the water-rich pores (see Figure 3c,d). Upon the first mechanical stimulation of the coacervates, which occurred 3 d into the release profile (i.e., when the swelling was the fastest), the compression treatment reduced the slope of the coacervate weight versus time curves (cf. red circles and blue diamonds in Figure 3a), while the perforation (grey squares in Figure 3a) did not produce a significant effect. At longer times, however—starting with the second stimulation (performed 6 d into the release experiment), the coacervate weights decreased after each stimulation step. These decreases remained sharp over ~2 weeks, but then gradually became more subtle (Figure 3a). Moreover, their timing and intensity coincided with the spikes in the release rate (Figure 2b), which—along with the aforementioned plumes of RhB dye that were evident during early-stage perforation—suggested that the accelerated release was stimulated by the rupture of the RhB-loaded pores.

### 3.2. Effect of Mechanical Stimulation Frequency

With more frequent (daily) stimulation, the release became more uniform with, at least when the eluted RhB was measured daily, smooth cumulative release profiles (Figure 4a). Since the first mechanical stimulation was applied after 1 d, the onset of accelerated release became evident after the second day. With the compression (whose application was identical to that in Figure 2), the daily release rate first increased more than tenfold, and then decreased, as the effect of further mechanical stimulation evidently became less pronounced. In contrast, the perforation effect—which was reduced to two needle jabs per treatment (from the five jabs used when stimulated every 3 d) to maintain a similar average rate of needle jabs to that in Figure 2—generated a sigmoidal release profile where the release rate increased over the first five stimulation steps and then gradually decreased (Figure 4). Though the release rate with this perforation procedure did not rise as sharply (and did not reach as high of a crest) as that achieved through compression, this milder mechanical stimulation kept the average release rate above 1 µg/d longer than the (more intense) daily compression. Though after the first two weeks the mechanical stimulation effect became less pronounced for both activation procedures, this effect remained measurable (with a multifold increase in the release rate) throughout the 20-d experiment (see inset in Figure 4b).

Like in the case with the less frequent stimulation (in Figure 2 and Figure 3), the accelerated release was correlated with the reductions in swelling produced by the daily stimulation, and the reduction in swelling became apparent earlier with the compression than with the perforation (Figure 5). This correlation again supported the view that the accelerated release reflected the rupture and collapse of the RhB-loaded pores. Moreover, it suggested that, besides accelerating the slow payload release, the mechanical stimulation (as shown by the images in Figure 3b and Figure 5b) provides a potential approach to preventing excessive coacervate swelling, which can cause practical complications such as the blocking of fluid flow over the coacervate. Indeed, this swelling reduction persisted even after the coacervate stimulation stopped. Though after the experiment the coacervates—regardless of the stimulation type or frequency—swelled significantly when left in the deionized water for several weeks without further perforation or compression, the swelling of the stimulated samples remained much lower than that of the control samples. This continued reduction in coacervate swelling may have stemmed from the stimulation-driven collapse/elimination of many of their pores. While a detailed analysis of these post-stimulation swelling effects was beyond the scope of this study, their presence was clear from the continued visual observation of the water-immersed samples (as seen in the photographs in Appendix A).

### 3.3. Effect of Release Media

Collectively, the findings in Figure 2, Figure 3, Figure 4 and Figure 5 suggest that the mechanical stimulation effect on the PAH/TPP coacervate release performance is (at least partially) caused by the mechanical stimulation impact on the coacervate swelling. This observation raises the question of whether the accelerated release effect requires coacervate swelling. Since PAH/TPP coacervate swelling is much lower in PBS (which also models physiological conditions) [27], some of the above release experiments were repeated in PBS. Consistent with previous work, the PBS greatly diminished the PAH/TPP coacervate swelling, such that—possibly due to its PAH-complexing/PAH coacervation-promoting phosphate ions [38,39]—the average coacervate weight increased by no more than 50% over 14 d (Figure 6a). Thus, while (like in the cases where the RhB was being released into deionized water) the mechanical activation produced a reduction in the swelling, the changes in swelling were significantly less pronounced than those in Figure 3 and Figure 5, regardless of the mechanical stimulation. Indeed, as illustrated in Figure 6b, the mechanical activation (despite some variability in the angle from which the coacervate-bearing tubes were photographed) had no impact on the visual appearance of the coacervates.

As expected, the diminished effect of the mechanical stimulation on the swelling was also reflected in its effect on the release profiles (Figure 7a,b). After the burst release in the first hour, the average release rates rapidly dropped to approximately 1 µg/d within 1 d of contact time and continued to fall thereafter. Though this further drop in release rate was initially inhibited upon the first mechanical stimulation (after 1 d of contact time), the magnitude and duration of this release-promoting effect were much smaller than that seen in deionized water (cf. Figure 4 and Figure 7). Unlike the sharp acceleration of the RhB release seen in deionized water (which increased the early day release rates by more than tenfold and remained substantial over weeks), both types of mechanical stimulation in PBS produced only a roughly 2–3× initial increase, which became less pronounced with time, and after a week these release rates from the mechanically stimulated coacervates became either only slightly (less than 2×) higher than or indistinguishable from the controls (Figure 7b). This reduction in the stimulation effect indicates that the mechanical stimulation is most effective in accelerating release from PAH/TPP coacervates under conditions that promote pronounced swelling.

### 3.4. Further Discussion

Overall, the above experiments suggest that mechanical stimulation of release from complex coacervate matrices works well in media where significant swelling occurs (exemplified by the deionized water in this work) but has limited efficacy under conditions that inhibit coacervate swelling (e.g., in PBS, when used with PAH/TPP coacervates). Given this inhibitory PBS effect, the mechanical stimulation will (when used with PAH/TPP coacervates) likely work best in applications without high phosphate ion concentrations (i.e., outside of physiological media). It may therefore be better-suited to household and industrial applications, where the coacervates might either be in contact with low-ionic-strength solutions or with salts that promote PAH/TPP coacervate swelling [27]. Under such conditions, the mechanical stimulation strategies explored herein can produce a multifold increase in release rates and—in cases where the duration of efficacious sustained release is limited by the release rate ultimately becoming too slow (e.g., in sustained disinfections [21])—could extend the duration of the desired effect.

More broadly, to optimize this mechanical stimulation effect, the coacervates should likely be prepared under conditions that promote their swelling (e.g., by mixing the complex coacervate-forming species in nonstoichiometric charge ratios [33]). Moreover, to prevent the loss of mechanical stimulation efficacy under physiological conditions (with phosphate contents akin to PBS), coacervates can also be prepared from polycations that (unlike PAH [38,39]) do not complex and undergo complex coacervation with phosphate ions. Provided that conditions for coacervate swelling and mechanically activated pore collapse can be achieved, the mechanical stimulation procedures reported herein provide an attractive potential strategy for accelerating release from complex coacervates when it (over time) becomes too slow. Besides being able to reactivate the sustained release once it slows down, this coacervate stimulation might be useful in situations where the active payload release from the coacervates is too slow to be efficacious from the start, or when a periodic/pulsatile dosing might be desired (e.g., for pulsatile drug delivery [40,41,42] or to periodically disinfect devices such as water shower heads/hoses [43,44] or dental unit waterlines [45,46] without continuously exposing their users to chemical biocides). Likewise, it may have applications outside of traditional sustained release applications, such as sensing (e.g., where a dye such as the RhB used in this work is released each time that a device is touched to generate a colorimetric signal).

## 4. Conclusions

We have shown that, when the sustained release of small molecules from complex coacervate matrices is slower than desired, it can be accelerated through mechanical stimulation, akin to flavor release from a piece of chewing gum. When this strategy is used with PAH/TPP coacervates, the release of small, water-soluble molecules into low-ionic-strength water can be greatly accelerated when the coacervate is periodically perforated or compressed. The release profile achieved through this strategy can be varied by tailoring the frequency and type (e.g., perforation versus compression) of the mechanical stimulation.

The accelerated release evidently stems from the rupture and subsequent collapse of payload solution-filled pores within the coacervate phase and is—at least under the conditions examined herein—most pronounced under conditions where there is significant (pore volume-enhancing) swelling. Besides increasing the overall dosing of the slowly releasing payload, the mechanical stimulation enables pulsatile release, where the rate increases sharply immediately upon stimulation and then returns to baseline levels, and (since the coacervate volume is reduced by the mechanical rupture/collapse of the pores) provides a pathway to moderate coacervate swelling in cases where it becomes excessive. Collectively, these findings show that mechanical stimulation of coacervate matrices provides a potential approach to overcoming insufficient release rate problems in their applications, such as sustained disinfection or drug release, and could open doors to new coacervate-based technologies.

## Figures and Tables

**Figure 1 polymers-15-00586-f001:**
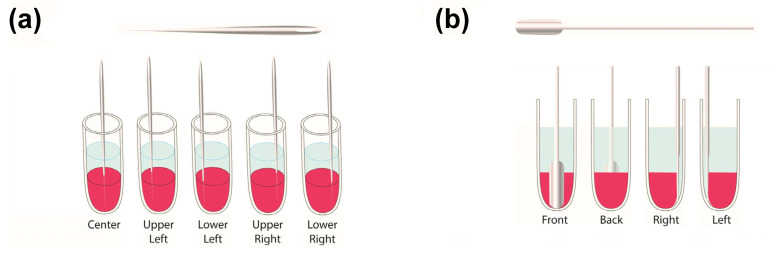
Mechanical stimulation schemes showing the approximate locations of the (**a**) perforations by the 5 needle jabs and (**b**) compressive spatula insertions. The perforated samples were jabbed multiple times during each treatment, while those subjected to compression were compressed once per treatment, either on the front, back, left, or right side of the sample.

**Figure 2 polymers-15-00586-f002:**
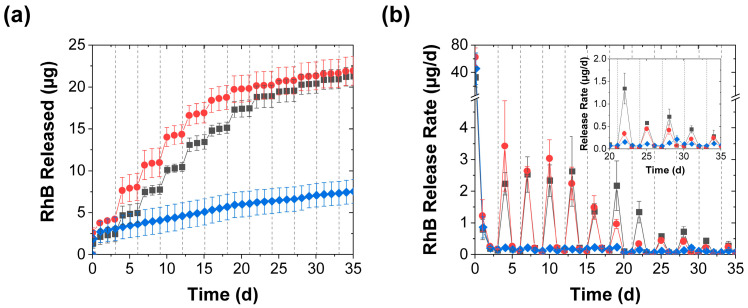
RhB release from coacervates into deionized water achieved with (■) periodic perforation by five needle jabs, (●) periodic compression, and (♦) without mechanical stimulation and shown in terms of both (**a**) the total RhB mass released and (**b**) the release rate (mean ± SD). The inset provides a closeup of the slower release rates near the end of the experiment. The coacervates in this experiment were stimulated every 3 d using either five needle jabs or a spatula. The solid lines are guides to the eye, while the dashed vertical lines mark the mechanical stimulation times.

**Figure 3 polymers-15-00586-f003:**
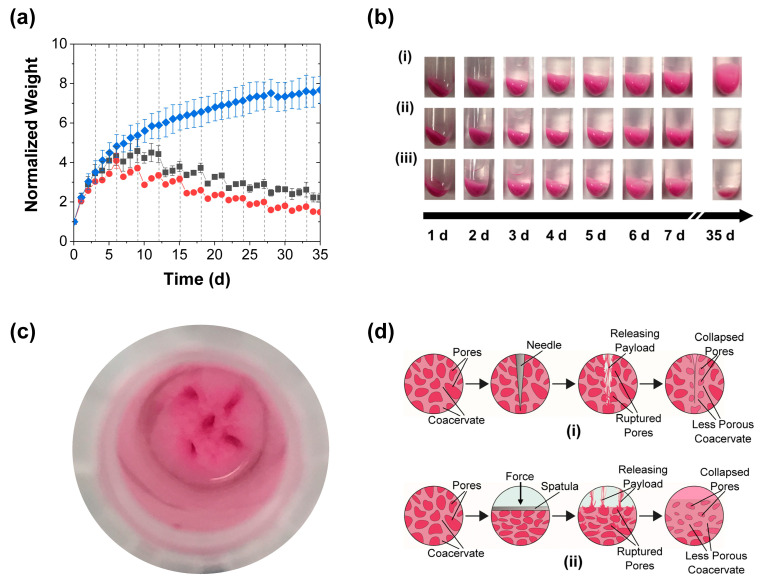
Coacervate swelling during the RhB release into deionized water achieved with (■) periodic perforation, (●) periodic compression, and (♦) without mechanical stimulation and characterized by (**a**) gravimetric analysis of the evolutions in normalized weights (mean ± SD) and (**b**) digital photography (i) without mechanical stimulation, (ii) with periodic perforation, and (iii) periodic compressions. Also shown are (**c**) a top view of a coacervate sample after a perforation treatment and (**d**) schemes of the solvent-filled pores collapsing after each mechanical treatment. The coacervates in this experiment were stimulated every 3 d using either five needle jabs or a spatula. All coacervate weights are normalized to their initial values at the start of the release experiment. The solid lines are guides to the eye, while the dashed vertical lines mark the mechanical stimulation times.

**Figure 4 polymers-15-00586-f004:**
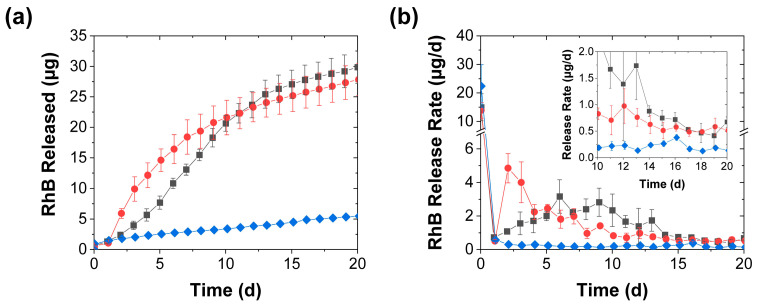
RhB release from coacervates into deionized water achieved with (■) daily perforation with two needle jabs, (●) daily compression, and (♦) without mechanical stimulation and shown in terms of both (**a**) the total RhB mass released and (**b**) the release rate (mean ± SD). The inset provides a closeup of the slower release rates near the end of the experiment, while the lines are guides to the eye.

**Figure 5 polymers-15-00586-f005:**
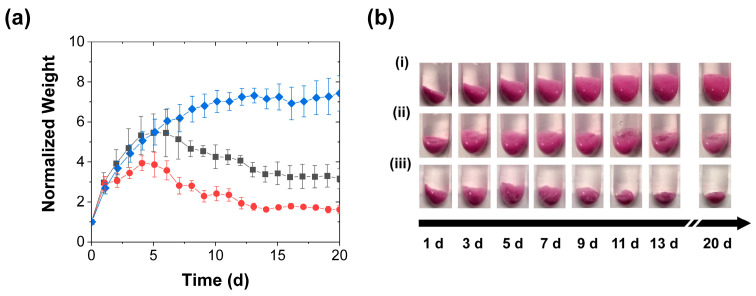
Swelling of coacervates during RhB release into deionized water achieved with (■) daily perforation by two needle jabs, (●) daily compression, and (♦) without mechanical stimulation and characterized by (**a**) gravimetric analysis of the evolutions in normalized weights (mean ± SD) and (**b**) digital photography (**i**) without mechanical stimulation, (**ii**) with periodic perforation, and (**iii**) periodic compression. All coacervate weights are normalized to their initial values at the start of the release experiment. The lines are guides to the eye.

**Figure 6 polymers-15-00586-f006:**
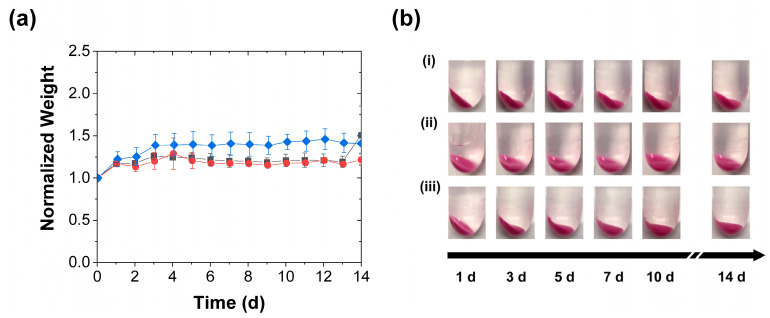
Coacervate swelling in 1 × PBS with (■) daily perforation with five needle jabs, (●) daily compression, and (♦) without mechanical stimulation and characterized by (**a**) gravimetric analysis of the evolutions in normalized weights (mean ± SD) and (**b**) digital photography (**i**) without mechanical stimulation, (**ii**) with periodic perforation, and (**iii**) with periodic compression. All coacervate weights are normalized to their initial values at the start of the release experiment. The lines are guides to the eye.

**Figure 7 polymers-15-00586-f007:**
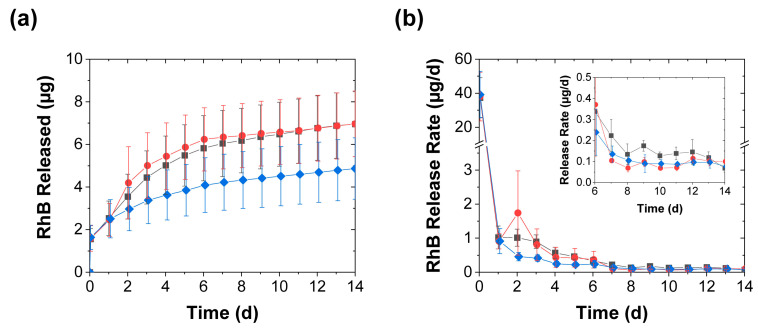
RhB release from coacervates into 1 × PBS achieved with (■) daily perforation with five needle jabs, (●) daily compression, and (♦) without mechanical stimulation and shown in terms of both (**a**) the total RhB mass released and (**b**) the release rate (mean ± SD). The inset provides a closeup of the slower release rates near the end of the experiment, while the lines are guides to the eye.

## Data Availability

The data presented in this study are available upon reasonable request from the corresponding author.

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
