# Peer review of "Accelerating Payload Release from Complex Coacervates through Mechanical Stimulation"

_polymers, 2023, doi:10.3390/polym15030586_

Round 1
Reviewer 1 Report
In this study, Hatem and Lapitskt experimentally study the mechanical stimulation (by needle perforation and compression) effects on the drug release behavior in a polyelectrolyte complex coacervate liquid. In aqueous solutions, significant improvement of release rate are observed by applying the two simulates, compared to the case when no actions are taken. The authors then related the improvement of the drug release to the reduction of the swelling of the coacervate upon applying the periodic stimuli (per 3 days and 1 day), which is evidenced by the controlled experiments of the same coacervate in an anti-swelling media.
The article is well written and organized. I enjoy reading this article and satisfy by its completeness, good scientific merit and well-design experiments. The conclusions are well-supported by the presented data. The application of coacervates is still a challenge field in the community and I am glad the authors explored the improvements on drug delivery. The article can be published as it is or with very minor revision. I only have a few comments for the authors' considerations:
1. The water pore interpretation is certainly informative. Presumably, if no significant structure damage, given enough time, the volume/weight of the coacervate will recover to swollen state after the stimuli. Did the author observe obvious volume recovery after stop the perforation/compression in water for a long time?
Reviewer 2 Report
The authors demonstrated the payload release acceleration effect of small molecules encapsulated in coacervates via mechanical stimulation, such as perforation and compression. These mechanical stimulations activate the rupture and collapse of solvent-filled pores and the environment that suits such effects. This study provided insight into efficient payload release. The authors provided detailed information on the experiment design and result analysis to show the acceleration effect when mechanical stimulation was applied periodically over a month. Thus, the reviewer only have minor questions as follows before recommending publishing:
To gain insight into the stimulation effect, the authors measure the weight of coacervate from time to time and replace the release media (water as the supernatant). At the same time, the coacervate surface was rinsed with water or PBS, then dry it with Kimwipe. How does this rinsing and drying step affect the RhB release? Usually, when rinsing or drying, external force unavoidably is applied to the coacervate. How does this force affect the release?
